# Histone Methylases and Demethylases Regulating Antagonistic Methyl Marks: Changes Occurring in Cancer

**DOI:** 10.3390/cells11071113

**Published:** 2022-03-25

**Authors:** Joyce Taylor-Papadimitriou, Joy M. Burchell

**Affiliations:** School of Cancer and Pharmaceutical Sciences, King’s College London, London SE1 9RT, UK; joy.burchell@kcl.ac.uk

**Keywords:** epigenetics, cancer, Polycomb, Trithorax, histone methylases, histone demethylases

## Abstract

Epigenetic regulation of gene expression is crucial to the determination of cell fate in development and differentiation, and the Polycomb (PcG) and Trithorax (TrxG) groups of proteins, acting antagonistically as complexes, play a major role in this regulation. Although originally identified in Drosophila, these complexes are conserved in evolution and the components are well defined in mammals. Each complex contains a protein with methylase activity (KMT), which can add methyl groups to a specific lysine in histone tails, histone 3 lysine 27 (H3K27), by PcG complexes, and H3K4 and H3K36 by TrxG complexes, creating transcriptionally repressive or active marks, respectively. Histone demethylases (KDMs), identified later, added a new dimension to histone methylation, and mutations or changes in levels of expression are seen in both methylases and demethylases and in components of the PcG and TrX complexes across a range of cancers. In this review, we focus on both methylases and demethylases governing the methylation state of the suppressive and active marks and consider their action and interaction in normal tissues and in cancer. A picture is emerging which indicates that the changes which occur in cancer during methylation of histone lysines can lead to repression of genes, including tumour suppressor genes, or to the activation of oncogenes. Methylases or demethylases, which are themselves tumour suppressors, are highly mutated. Novel targets for cancer therapy have been identified and a methylase (KMT6A/EZH2), which produces the repressive H3K27me3 mark, and a demethylase (KDM1A/LSD1), which demethylates the active H3K4me2 mark, are now under clinical evaluation.

## 1. Introduction

Epigenetic regulation of gene expression plays a crucial role in the determination of cell phenotype and the functions of the Polycomb (PcG) and Trithorax (TrxG) groups of proteins, acting antagonistically, are core to the determination of cell fate in development and differentiation. In 1947, working with Drosophila mutants, the first Polycomb gene (PC) was identified [1] and twenty years later the aberrant positioning of embryonic segments which were caused by mutations of PC were documented and related to the ectopic expression of *Homeotic* genes (*Hox*) [2]. Identification of other genes showing the same phenotype upon mutation led to the specification of the Polycomb group of proteins. Subsequently, *Trithorax* was identified as a gene which suppressed the changes induced by Polycomb [3], and identification of other genes with the same function as *Trithorax* defined the Trithorax group of proteins. It is now clear that the PcG and TrxG complexes, acting antagonistically, play a central role in sustaining a balanced state of gene expression in numerous cellular processes.

It was later found that these complexes contain histone methylases [4], which can repress or activate transcription depending on the specific lysine being methylated. Thus, trimethylation of lysine 27 on histone 3 (H3K27me3) by the Polycomb complexes can repress gene transcription, including that of *Hox* genes [5], while methylation by Trithorax complexes at lysine 4 (H3K4me2/3) and lysine 36 (H3K36me2/3) can activate transcription [6,7,8]. The Polycomb/Trithorax system is evolutionarily conserved and the components of these complexes which act antagonistically in mammalian development and differentiation are now largely defined [9,10,11]. There are two main sub-groups of TrxG, which generally promote an active chromatin state, the histone lysine methylases which, when methylating K4, act as components of the COMPASS family of complexes, and the SWI/SNF family, which are involved in ATP-dependent chromatin remodelling [12]. While the original Trithorax complexes with their respective methylases methylate K4, the methylases which methylate K36, producing the K36me2/3 active marks also antagonise Polycomb and regulate *Hox* genes [7]. One of these methylases (ASH1L) has been formally recognised as Trithorax, acting in a complex [8]. We will therefore in this review include the methyl transferases, which add methyl groups to K36 on histone 3, as well as those which methylate H3K4 and H3K27.

Enzymes which demethylate methylated histones were discovered more recently, the first being KDM1/LSD1 (lysine demethylase 1) [13]. LSD1 and LSD2 are amine oxidases which can demethylate H3K4me1/me2, while the other demethylases identified later use the JmjC domain to catalyse demethylation [14]. Unlike some of the methylases, including the KMT2 and KMT6 enzymes, which add methyl groups to K4 and K27, respectively, the histone lysine demethylases (KDMs) do not operate from specific complexes but are recruited to chromatin in various ways [15,16,17]. With the exception of ASH1L, which does operate from a complex, this is also true for the methylases which methylate H3K36 to produce H3K36me2 and H3K36me3.

The PcG and TrxG complexes and their component histone methylases have been the subject of several reviews [9,10] and separate reviews covering the histone demethylases are also available [15,16,17]. Here we review both the methylases and demethylases and their interaction in governing the methylation state of the H3K27me3 suppressive mark, and of the methylated H3K4 and H3K36 active marks, and describe the changes occurring in these interactions in cancer. We also focus on any enzyme independent actions that are relevant to cancer (see Box 1). Incorporating the methylases and demethylases for H3K36 has allowed a discussion of the important interaction of the KDM2B demethylase with a Polycomb complex. Figure 1 shows the products synthesized by the KMTs and the substrate specificity of the KDMs acting on lysines 4, 27 and 36 on histone 3.

Box 1Enzyme dependent and independent function.The KMT and KDM proteins have other functional domains in addition to their catalytic function. For example, acting independently of its enzyme activity, KDM6A has been shown to recruit the KTM2D complex and a histone acetyl trans-ferase to chromatin to establish active enhancers. Since some of these proteins are being considered as targets for cancer therapy and inhibitors under development largely target the enzyme activity, we also focus on the relevance of enzyme activity to the changes occurring in carcinogenesis [18].

## 2. Regulation of Methylation of Lysine 27 on Histone 3

### 2.1. The PRC1 and PRC2 Complexes Methylating H3K27

In mammals, there are two Polycomb complexes: PRC1 and PRC2 (Polycomb Repressive Complexes 1 and 2). However, within these classes are subclasses, which differ according to the profile of the component proteins and/or the proteins they associate with [10,19,20]. Both PRC1 and PRC2 are required for repressive function. The methylase (KMT6A/EZH2) found in PRC2 complexes generates H3K27me3, while PRC1 complexes ubiquitinate lysine 119 on histone 2A. The pathway used to recruit the two complexes can be different, where the first complex recruited can be a PRC1 or a PRC2 complex as shown in Figure 2 [10,19,20,21]. However, recruitment of either complex to chromatin is not achieved directly by the core components of the complexes, but by associated proteins which can bind to unmethylated CpG-rich sequences.

*PRC1 complexes*: There are multiple PRC1 complexes in mammals. They are broadly classified as canonical or non-canonical, although all the complexes contain either RING1A or RING1B, which ubiquitinate H2A histone (H2AK119ub1), and one of the six PCGF proteins (Polycomb group really interesting new gene (RING) finger 1–6 paralogues).

The canonical complexes contain one of the chromodomain-containing proteins (CBX 2, 4, 6, or 8), which bind to H3K27me3, while the non-canonical complexes contain either RYBP (RING 1/YY1 binding protein) or its homologue, YAF2. The canonical complexes are only associated with PCGF2 or 4 and can be recruited following production of H3K27me3 by PRC2.1 [10,19,21,22,23] and see Figure 2 left hand pathway.

Some non-canonical PRC1 complexes can be recruited directly to chromatin, depending on the associated proteins. The ncPRC1.1 complex, containing PCGF1 and the H3K36me2 histone demethylase KDM2B, is recruited globally to unmethylated CpG islands by KDM2B, through its CxxC-ZF domain, independently of its enzyme activity [24,25,26,27,28]. KDM2B is also required for ubiquitination by RING1B and recruitment of PRC2.2 (see below) depends on this ubiquitination by PRC1.1 (see Figure 2 right hand pathway) [21,26,27,28,29,30,31,32]. PRC2.2 produces the H3K27me mark recognised by the CBX protein in a cPRC1 complex, which can be recruited to enhance ubiquitination [20,21,32].

*PRC2 complexes:* The KMT6A/EZH2 and KMT6B/EZH1 (enhancer of zest homologues1 and 2) methylases, found in PRC2 complexes, add the trimethyl group to lysine 27 on histone 3, creating the H3K27me3 suppressive mark [5]. In the complete PRC2 complexes, EZH2 can methylate H3K27 effectively and is expressed in the developing embryo and in proliferating cells, while the EZH1 PRC2 complex has low levels of methylase activity and is present in non-dividing cells in the adult [19]. However, the domain structure of the two KMT6 methylases is similar, and similar to all the methylases they include the catalytic SET domain (Su(var)3-9, Enhancer-of-zeste and Trithorax).

The SUZ12, EED, and EZH2 core components are present in stoichiometric amounts in the PRC2 complexes [19,20,21] and this core complex associates with one or more of several non-core components in sub-stoichiometric amounts to form complexes classed as PRC2.1 or PRC2.2 [21,33,34,35]. PRC2.1 complexes, through an interacting PCL protein, can be recruited to unmethylated CpG islands, thus initiating the pathway illustrated in Figure 2 (left hand pathway) which recruits cPRC1. The binding of the ncPRC1 complex described above recruits PRC2.2 complexes (Figure 2, right hand lane).

PRC2.1 and PRC2.2 are found together globally at the same genes, suggesting that the two recruitment pathways are coordinated.

### 2.2. Demethylation of H3K27me3

The two histone demethylases KDM6A/UTX (ubiquitously transcribed tetratricopeptide repeat, X chromosome) and KDM6B/JMJD3 (Jumonji Domain Containing 3) catalyse the demethylation of the repressive H3K27me3 mark, and both proteins play an important role in embryonic development and in regulation of *Hox* genes [36]. However, some functions operative in the early embryo are independent of the demethylase activity of these KDMs [37].

*KDM6A/UTX and UTY*: The gene coding for KDM6A is on the X chromosome and not subject to gene silencing. A gene on chromosome Y (UTY) codes for a protein which is similar to UTX but, because of mutations in the JmjC domain, it has no demethylase activity. Homozygous knock out of UTX leads to embryonic lethality in females, but males survive to term, indicating that a demethylase-independent function of UTY (probably also carried out by UTX) is important for embryonic development and survival [37]. As stated earlier, KDM6A/UTX has been studied as a component of the KTM2C and KTM2D trithorax complexes, playing a major role in establishing active enhancers carrying the H3K4me1 and H3K27Ac marks independently of its enzyme activity [18].

## 3. K27 Methylases and Demethylases in Cancer

### 3.1. The EZH2 Methylase: Expression, Mutations, and a Therapeutic Target

Both increases and decreases in H3K27 methylation are found in cancer, and EZH2 can act as an activator of transcription independently of Polycomb (see Figure 3).

Activating mutations in the SET domain of EZH2 (KMT6A) are seen in some lymphomas (see Table 1) and levels of EZH2 are increased in many carcinomas, including breast and prostate, and in multiple myeloma, where high expression is associated with poor prognosis [38,39,40].

The effects of loss-of-function mutations on other epigenetic factors which oppose EZH2 activity can also result in increased function of EZH2. Inactivating mutations affecting the enzyme activity of KDM6A/UTX [41], or of components of the SWI/SNF Trithorax complexes, which normally oppose Polycomb activity, are very common, and these mutations can confer sensitivity to EZH2 therapy ([42,43,44], see Figure 3). The action of the SWI/SNF mutations in cancers relies on both methylase-dependent and independent effects, and the independent effects may involve stabilisation of the PRC2 complex [43].

Inactivating mutations of EZH2 are seen in myeloid disorders [45,46], while in 25% of T-ALL (T cell Acute Lymphoblastic Leukaemia), inactivating mutations are found, not only in *EZH2,* but also in *SUZ12* [47]. These mutations are accompanied by activating mutations of *NOTCH1,* which opposes EZH2, indicating a Notch:PRC2 pathway interaction in tumorigenesis in TLL.

That EZH2 can act as an oncogene or as a tumour suppressor is seen in AML, where EZH2 expression is required for maintenance of the disease but acts as a tumour suppressor at the induction stage [48,49].

*EZH2 acting independently of the Polycomb complex*; EZH2 can activate expression of the androgen receptor (AR) independently of the PRC2 complex in methylase-dependent and independent ways and thus increases the expression of AR target genes ([50,51] see Figure 3).

*Inhibitors of EZH2:* Small molecular weight inhibitors of the EZH2 methylase activity have been developed and many Phase I/II trials are now ongoing to evaluate their efficacy against solid cancers, including castrate-resistant prostate cancer (in combination with an inhibitor of AR signaling), as well as cancers of the hemopoietic system. The inhibitor *Tazemetostat* has been most widely used and has FDA approval for treatment of patients with two types of follicular lymphomas, which can show activating mutations of EZH2 [52,53].

**Table 1 cells-11-01113-t001:** Mutations and translocations of methylases and demethylases in cancer.

*Type of Mutation or Translocation*	*Gene*	*Cancer Type*	*References*
**Inactivating** Commonly found and high in bladder cancer	***UTX/KDM6A*** Affecting H3K27 methylation and activation of enhancers	Common in T-ALL	[41]
	***KMT2C and KMT2D*** Affecting H3K4 mono-methylation at enhancers.	Bladder cancer ccRenal Carciomas non-Hodgkin’s lymphoma neuro-ectodermal tumours	[54,55,56]
	***SETD2/KMT3A*** Inhibiting synthesis of H3K36me3 and therefore DNA damage response	ccRenal Carcinoma 32% Enteropathy-associated T cell lymphoma (EATL)	[47,54,57,58,59]
	***EZH2/KMT6A*** Affects H3K27 methylation	Acute T cell leukaemia and myeloid disorders	[40,45,46,47]
**Activating mutations**	***EZH2/KMT6A*** Enhancing EZH2 function	Lymphomas	[48]
	***NSD2/KMT3B* p.E1099K** Increased enzyme activity	10% of all ALL and especially paediatric B-ALL	[60,61]
**Duplication**	***MLL1/KMT2A*** partial tandem duplication (PTD) of exons 5 to 11	4–7% of AML	[62]
**Translocations making fusion proteins** Found in specific cancers	** *KMT2A-with 3′ domain from multiple genes including SEC components* **	70% of infant leukaemia (ALL and AML)	[63,64]
	** *NUP98-NSD1/KMT3B* **	5% of AML	[65]
	** *NUP98-NSD3/KMT3F* **	AML (rare)	[65]
	***NSD2 placed under the control of the strong IGH intronic Em enhancer t* [4,5,6,7,8,9,10,11,12,13,14]**	15–20% Multiple myeloma 5–10% Paediatric ALL	[58,60,66]
	**SX18/SSX**	90% Synovial sarcomas	[67]
**Mutations in non-canonical histones** Found in specific cancers	***H3F3A*** K27K to M G34 to R/V/D G34 to W/V	Childhood glioblastomas >90% Giant Cell tumours	[68,69,70,71]
	***H3F3B*** K36 to Methionine	>90% Chondroblastoma	[68]

Inhibitors destabilizing PRC2 are also available and cell lines harboring a SWI/SNF mutation, which were not sensitive to inhibition by an EZH2 methylase inhibitor, were inhibited by a compound that blocks the EZH2 interaction with EED (see Figure 2 and Figure 3 and Box 2), and results in breakdown of the complex [48]. Therefore, demethylase-independent functions of EZH2 also contribute to its oncogenic effect. Moreover, EZH2, and EED interact directly with DNA methylases [72] and with HDACs [73], and combined targeting of EZH2 and HDACs has been recommended.

*Interaction of EZH2 with HOTAIR*: The EZH2 Polycomb complex can interact with the long non-coding RNA HOTAIR, which is highly expressed in breast cancer metastases, and a small molecular weight inhibitor of this interaction has been developed [74].

### 3.2. Mutations in Cancer Converting H3K27 to Methionine

Mutations occur in cancer in the non-canonical histone genes H3F3A or H3F3B, coding for histones 3.3 A and B [68], which are recruited to chromatin independently of mitosis by the ATRX/DAXX complex. The most striking mutation affecting Polycomb function is the change of lysine 27 in histone H3F3A to methionine, and is seen in 70% of paediatric glioblastomas ([69,70] and see Table 1). Moreover, although the mutation occurs in only one allele of H3F3A, the methylase activity of EZH2 is globally inhibited, resulting in considerably reduced levels of H3K27me3 [71]. However, the K27 methylase activity (although reduced) is required by the cancer cells to inhibit neural differentiation and allow proliferation. Therefore, EZH2 has been proposed as a target for therapy of these cancers [75].

### 3.3. KDM6A and KDM6B Demethylases in Cancer

The demethylase KDM6A/UTX is frequently mutated in cancer [41] and these mutations are not confined to the JmjC domain, so that demethylase-independent effects are possible (Table 1 and Table 2). In such examples, UTY could be functional, but UTY mutations can also occur, and complete loss of the Y chromosome is not uncommon in males with a KDM6A mutation in cancer [41]. Where the mutation affects the demethylase activity of UTX, gender specific effects are seen.

*T Cell Acute Lymphoblastic Leukaemia (T-ALL) and Acute Myeloid Leukaemia (AML):* The role of both demethylases has been widely studied in T-ALL leukaemia, where they play opposing roles: KDM6A (UTX) acts as a tumour repressor, and mutations of KDM6A are seen dominantly in male patients, where there is a gender bias of 3:1 (male:female) in development of T-ALL (see Table 1). Moreover, ablation of KDM6A can lead to development of AML in specific mouse models [76]. KDM6B, on the other hand, interacts with NOTCH (which presents with activating mutations in T-ALL) in co-activating target genes, including oncogenes. Inhibition of KDM6B demethylase activity has been suggested as a possible therapy for this type of leukaemia [77] and the BET inhibitors, which inhibit myc expression, could be effective (see later section on K4 methylation).

*Multiple myeloma (MM):* In MM, loss of KDM6A resulted in increased proliferation, expression of c-myc, and loss of expression of E-cadherin [44]. Inhibition of EZH2 was effective in reversing these effects, suggesting that loss of UTX leads to abnormal PRC2-mediated repression, and that inhibition of EZH2 could be effective in MM.

*Carcinomas:* Work with pancreatic cancer (PANCS) supports the concept that KDM6A normally acts as a tumour suppressor, since ablation of KDM6A can lead to development of PANCS in a mouse model, and in patients, oncogenes are activated in females and in males with UTY loss [78]. In breast cancer, KDM6A is reported to inhibit the epithelial-to-mesenchymal transition (EMT) by silencing the expression of EMT transcription factors in collaboration with LSD1 and HDAC1 [79].

Box 2Targeting EZH2 in cancer.Clinical trials targeting EZH2 methylase (KMT6A) activity for cancer therapy are ongoing. Pre-clinical studies indicate that inhibiting interactions of compo-nents of PRC2 to destabilise the complex could provide an alternative targeting strategy.The mutation of lysine 27 to methionine in a non-canonical histone is seen in 70% of paediatric gliomas and inhibitors of EZH2 or BET proteins are therapeu-tic options.The main body of data indicates that the KDM6A/UTX demethylase protein acts as a tumour suppressor, and that tumours lacking or carrying mutations in KDM6A may, in some cancers, be targeted with either EZH2 or BET inhibitors.

**Table 2 cells-11-01113-t002:** Methylases and demethylases: functions independent of the enzyme activity of the two classes of enzymes.

*Functions in Normal Mouse Embryo Development and Viability Independent of Enzyme Activity*	*References*	*Enzyme Independent Effects in Cancer*	*References*
***KMT2A/MLL1*** Progeny from mating of KMT2A heterozygous methylase null mice are produced at the expected Mendelian ratios.	[80]	***EZH2/KMT6A*** Can activate transcription of genes targeted by the androgen receptor (AR) independently of the Polycomb complex which is required for methylase activity (Figure 3).	[50]
***KDM6A(UTX) and UTY*** Male mice mutant for UTX can survive with UTY even though it has no demethylase activity.	[37]	***LSD1/KDM1A*** Interaction of LSD1 with transcription factors activates expression of a network of genes favouring growth of CRPC prostate cancers (Figure 3).	[81,82]
***KDM2B*** The ncPRC1.1 complex is recruited to chromatin in mouse ESCs by KDM2B through its CxxC-zinc finger (CxxC-ZF) domain, independently of its demethylase activity.	[25,26]	***KDM2B*** **Synovial Cancer**: KDM2B is required for proliferation independently of demethylase activity (>90% of these cancers). **Pancreatic Cancer:** KDM2B can activate genes involved in ribosomal and mitochondrial function.	[67] [83,84]
***KDM5B*** The ΔARID mice expressing KDM5B with no demethylase activity are viable and fertile.	[85,86]	***KDM6A/UTX*** Mutations in domains other than the SET domain are seen in cancer suggesting a role of these domains in the tumour suppressor function.	[41]

## 4. Regulation of Methylation of Lysine 36 on Histone 3

### 4.1. H3K36 Methylation: Methylases Associated with SET2 Domain

Methylation of H3K36 opposes Polycomb repression [87,88,89]. In mammals, there are several enzymes that can dimethylate H3K36 but only one (KMT3A/SETD2) can add the third methyl group, even though all contain the SET2 domain originally identified in the yeast SET2 methylase [90].

*Trimethylation of H3K36***:** SETD2/KMT3A is the only enzyme which can trimethylate H3K36 [90,91] and requires the H3K36me2 substrate formed by the dimethylases [92,93,94]. H3K36me3 regulates homologous recombination (HR), non-homologous end joining (NHEJ), as well as mismatch repair (MMR) [95]. Therefore, by producing the H3K36me3 mark, SETD2 protects the DNA damage response (DDR) and also regulates transcriptional elongation and splicing ([96] and see below).

*Enzymes dimethylating H3K36:* The NSD (nuclear receptor SET domain-containing) family of enzymes (NSD1 (KMT3B), NSD2 (KMT3G/F), and NSD3 (KMT3G/F)) are largely responsible for the global dimethylation of H3K36, see Figure 1. The NSD1 protein (KMT3B), has been widely studied [92] and is required for embryonic development. Mutations in NSD1 are responsible for the childhood overgrowth condition known as Sotos syndrome and for some cases of Weaver syndrome [97,98]. Where the NSD proteins have been studied in disease, no reference is given to them being members of a specific complex, but these proteins do regulate *Hox* genes [89]. While the ASH1L (KMT2H) methylase, which does operate in a trithrorax complex, also dimethylates K36, it is more selective than the NSD methylases, but the genes targeted include *Hox* genes [99,100].

*Methylation of H3K36 and transcriptional elongation*: In mammals, transcriptional elongation is accompanied by H3K36me3 deposition by SETD2 (KMT3A) towards the 3′ end of transcription. The H3K36me3 mark recruits the Dnmt3b DNA methylase, which protects the gene body from cryptic transcription initiation [101]. Thus, the trimethylation of H3K36 ensures the fidelity of gene transcription initiation, as well as protects the function of the DDR [90,101].

### 4.2. Demethylation of Methylated H3K36

The two demethylase families which remove methyl groups from methylated H3K36 are KDM2 and KDM4.

*KDM2A and KDM2B demethylases active on H3K36me2:* KDM2A/(FBXL11) and KDM2B (FBXL10) specifically demethylate H3K36me2 and H3K36me1. KDM2A was the first Jumonji domain containing demethylase to be isolated and characterised [14], and KDM2B has been clearly shown to have the same activity by the Zhang laboratory [24]. Genetic ablation of KDM2B in mice results in early embryonic lethality, and the biological effects of KDM2B have been widely studied in mouse embryo fibroblasts (MEFs), and in mouse embryo stem cells (mESCs) [102,103]. Over expression of KDM2B results in immortalisation of MEFs, by repressing the Ink4a/Arf locus [24,104,105] and this effect is dependent on the demethylase activity.

*KDM2B and the Polycomb complexes:* As a component of the PRC1.1 Polycomb complex, KDM2B recruits the complex to unmethylated CpGs and through ubiquitination of histone 2A (H2AK119ub1), PRC2.2 is recruited. Studies with mESCs show that both PRC1 and PRC2 complexes can interact with KDM2B. However, while the actual recruitment of the PRC1.1 complex to chromatin by KDM2B is not dependent on the demethylase activity, the enzyme activity could be involved in other functions stimulated by these interactions. (See section on Polycomb complexes and Figure 2).

*The KDM4 demethylases active on H3K36me3:* The KDM4 demethylases KDM4A, B and C demethylate H3K36me3 but not H3K4me2, and can also demethylate H3K9me2/me3 [106,107]. The JmjN and JmjC domains are separated in the gene, but these domains lie adjacent in the protein and both domains are required for catalysis [108]. There appear to be no important functional domains in the sequence between the JmjN and C domains, in contrast to the KDM5 proteins, where functional domains are found to be separating JmjN and JmjC (see below). There is some functional redundancy in the KDM4A/KDM4C demethylases, as shown by KO experiments in mice [109].

## 5. Changes in Cancer Related to H3K36 Methylases

### 5.1. SETD2 (KMT3A): A Tumour Suppressor

Enteropathy-associated T cell lymphoma (EATL) is an aggressive, often lethal cancer, which can arise from celiac disease, and mutations in SETD2 are found in 32% of cases [57], while in clear cell renal carcinomas (ccRCC), deletions or mutations of SETD2 are also found [54,110,111]. Reduced levels of SETD2 are also seen in breast cancers and relate to poor prognosis [112]. The fact that SETD2 binds to p53 and is involved in p53 regulation of its downstream genes could relate to the function of SETD2 as a tumour suppressor [113].

### 5.2. Translocations and Mutations of NSD (KMT3B, G, F) Methylases

There are many reports of changes in the expression of the NSD methylases in a range of cancers, including lung, prostate, and breast cancer [65,66]. However, mutations and translocations are found in cancers deriving from haemopoietic cells.

*The E1099K mutations:* In 10% of ALL, and especially in paediatric B-ALL (B-precursor ALL), a mutation of glutamic acid to lysine occurs in the SET domain of NSD2, resulting in increased activity of the enzyme, leading to increased levels of H3K36me2 and decreased levels of H3K27 trimethylation [60,61], see Table 1.

*Translocations of the NSD methylases:* In 15–20% of multiple myeloma patients, a t(4:14) translocation places NSD2 under the control of the strong IGH enhancer (see Table 1), and this translocation is also seen in ALL [60]. Through increased activity of the NSD2 methylase, this results in the loss of gene-specific H3K36me2 modifications, and this disruption of organised H3K36me2 marks leads to the expression of genes driving the oncogenic programme [58,66].

In addition, some AML patients exhibit translocations of NSD1 (5–10%) or NSD3 (rare) [t(5;11)(q35;p15.5)], which lead to the production of a fusion protein (see Table 1). The 5′ component is taken from NUP98 (nucleoporin 98), which can bind to a histone acetylase (CBP/300) and is fused to a carboxy sequence from NSD1 or NSD3, which retains the dimethylase active domain [114,115]. The expression of the NUP98-NSD1 fusion protein in myeloid stem cells leads to their continued proliferation, inhibition of differentiation, and to the expression of *HOXA* genes (*A7*, *A9*, *A10*) which are normally repressed by Polycomb [89].

### 5.3. ASH1L in Leukaemia

The synergistic action of two methylases: KMT2A/MLL1 (methylates H3K4, see next section) and ASH1L (KMT2H), is associated with leukemogenesis and with regulation of *HOX* gene expression in leukaemias with KMT2A translocations [116]. Therefore, ASH1L has been proposed as a possible therapeutic target in leukaemia. Very recently a small molecular weight inhibitor has been developed which not only effectively inhibits the growth of cells from leukaemias with a translocation of KMT2A but also reduces the expression of the fusion protein target genes [117]. ASH1L operates as a component of a Trithorax complex (currently termed hAMC or dAMC), and another component of AMC, the MRG15 protein, binds to ASH1L and liberates an auto-inhibitory loop to activate the enzymic activity of the SET domain [118,119]. The new inhibitor binds next to the auto-inhibitory loop and blocks the activation of enzymic activity. This will be very useful for studying the specific functions of ASH1L as it shows >100 selectivity for ASH1L when compared to the other H3K36 dimethylases.

### 5.4. Mutations in Non-Canonical Histones Affecting K36 Methylation

As previously discussed, mutations in *H3F3A* and *H3F3B* (coding for H3.3A or H3.3B) occur in cancer and mutations which affect H3K36 methylation are found in childhood glioblastomas, and in giant cell tumours and chondroblastomas, both of which develop in young adults (see Table 1).

*Mutations in Glycine 34:* In childhood glioblastomas, the mutations of glycine 34 in H3.3A (G34/RVD) [68,69], replacing glycine with an amino acid with extended side chains, inhibits the action of SETD2, and the lack of H3K36me3 results in the blocking of mismatched repair (MMR), leading to genome instability [120,121]. Moreover, the mutated histone inhibits the function of KDM4 demethylases [122]. Expressing the G/R mutant in mouse ES cells results in changes in chromatin and the transcriptome, which mimic the triple KO of the three KDM4 proteins, indicating a global effect of this mutation in one allele of the *H3f3A* histone.

Mutations of glycine 34 in H3.3A, affecting H3K36 methylation, are also found in over 90% of giant cell tumours, a rare tumour of osteoblast lineage where progression to malignancy is uncommon [123].

*The mutation of H3K36 to methionine:* In chondroblastomas, a rare bone tumour developing in the second decade of life, the driver mutation converting lysine 36 to methionine in histone 3.3B has been reported in 95% of cases [123]. The K36M mutation leads to a global inhibition of the methylation of H3K36, loss of H3K36me3, and a gain and redistribution of K27me3, blocking differentiation of mesenchymal progenitor cells [124]. It is proposed that the mutations affecting methylation of H3K36 lead to increased activity of PRC2 in silencing genes required for cell differentiation.

The mutations which result in inhibition of the function of the tumour suppressor SETD2 (KMT3A), thus inhibiting the production of the H3K36me3 mark, block DNA repair and create genomic instability. Inactivating mutations in ccRCC are also found in two histone demethylases known to be tumour suppressors: KDM5C (demethylates H3K4me3) and KDM6A /UTX [54]. The data illustrate the prevalence of mutations of tumour suppressors in cancer, as well as the interplay of the histone methylases and demethylases. 

## 6. Changes in Cancer Related to H3K36 Demethylases

### 6.1. KDM2B Over-Expression in Cancer

This demethylase is highly expressed in lymphocytic leukaemia and in several carcinomas, including pancreatic, ovarian and breast, as well as in gliomas and synovial sarcomas [83,125].

*Acute Lymphoblastic Leukaemia (AML):* As seen in normal mouse and human haemopoietic stem and progenitor cells (HSPC), KDM2B is highly expressed in lymphoblastic leukaemic stem cells, and is required for their maintenance. Moreover, ectopic expression of KDM2B can transform HSPCs. Expression of KDM2B also enhances lineage commitment to T-ALL [103,126] where, working with Polycomb EZH2, developmental genes are repressed, through silencing of *p15^Ink4b^* [126].

*Pancreatic cancer:* High expression of KDM2B is seen in poorly differentiated pancreatic cancers (PANCS) and their metastases. Genomic analysis documenting genes affected by KDM2B expression and co-occupancy of promoters shows that KDM2B is associated with Polycomb in suppressing lineage-specific genes, and repression depends on the demethylase activity. In parallel, KDM2B, in association with KDM5A (demethylates H3K4me2/me3 see next section) and Myc, activates expression of genes involved in ribosomal and mitochondrial function [84]. The activation of genes by KDM2B is demethylase-independent (see Table 2).

*Glioblastoma (GBM*): KDM2B plays a major role in the maintenance of glioblastoma cancer stem cells (GSCs) and knockdown of KDM2B induces apoptosis and DNA damage [127]. The inhibitor GSK-J4 was found to decrease glioblastoma cell viability, as well as the self-renewal capacity of GSCs, and was said to be specific for KDM6A and B. However, it was later found not to be specific for demethylation of H3K27 [128]. Some of the inhibitory effect can be attributed to inhibition of the demethylase activity of KDM2B, since in GSK-J4 treated cells, levels of KDM2B were decreased and levels of H3K36me2, increased.

*Synovial sarcoma*: Synovial sarcomas are aggressive soft tissue sarcomas which affect children and young adults. They are often unresponsive to chemotherapy and can be lethal. Virtually 100% of synovial sarcomas express a fusion protein SX18/SSX. SS18 is a member of the human SWI/SNF chromatin remodeling complex and the SSX1, 2, and 4 proteins are transcriptional repressors interacting with Polycomb. KDM2B is required for proliferation of synovial sarcoma cells and recruits the fusion protein to chromatin [67], an action independent of its demethylase activity, to activate an aberrant differentiation programme normally subject to Polycomb repression (see Table 1 and Table 2).

### 6.2. KDM4 A–C Proteins Demethylating H3K36me3

Over expression of one or more of the KDM4 demethylases are found in many carcinomas, in gliomas, and in head and neck cancers [106]. Since the H3K36me3 mark plays multiple roles affecting the chromatin state and gene expression, therapeutic targeting of a specific H3K36me3 demethylase, which removes this mark, has been recommended for treatment of AML [129,130,131]. However, if the inhibitors target the demethylase activity, they would inhibit all the KDM4 proteins including those which only demethylate H3K9, and targeting demethylation of H3K9me2/3, as well as H3K36me3, could lead to side-effects in clinical application.

There are considerable data indicating that the KDM2B protein plays an important role in oncogenesis in a wide range of cancers, through maintenance of stem cell properties, inhibition of differentiation pathways and activation of oncogenic pathways. Suppression of developmental pathways appears to be dependent on the function of demethylase activity, while activation of oncogenic pathways can be independent of this activity.

## 7. Regulation of Methylation of Lysine 4 on Histone 3

There are many reviews on the enzymes that methylate and demethylate H3K4 and the reader is directed to references [9,10,15,16,55,132] as examples. Here we focus on the more recent developments and the association of changes in H3K4 methylation with cancer.

### 7.1. Methylases in COMPASS Complexes (Complex Proteins Associated with SET1)

COMPASS complexes methylate H3K4 and have evolved from the SET1 complex in yeast, to six complexes found in mammals [133]. The six methylases in humans are designated KMT2A, B, C, D, F, and G and have been also referred to as MLL1-4 (KMT2A-D) and SET1A and SET1B (KMT2F and G), see Figure 1. This nomenclature arose as the MLL1 protein (Mixed Lineage Leukaemia protein1), which undergoes translocation to form fusion proteins in a group of leukaemias, was found to be an analogue of Drosophila Trithorax (Trx) [134]. In the literature, the nomenclature for MLL2 and MLL4 has been used interchangeably and it is important to look for gene details and use the KMT nomenclature [55].

While all six methylases can methylate H3K4, different functions have been assigned to the different methylases [18,135,136], see Figure 1.

Homozygous deletion of any one of the KMT2 methylases in mice results in embryonic lethality, each showing unique defects in embryonic development [11]. However, transgenic mice homozygous for expression of a methylase null mutant of the *Kmt2a* gene are viable, with some defects in formation of the spinal column and in *Hox* gene expression. Evidently, functions other than the catalytic SET domain of KMT2A/MLL1 are also important for embryonic development ([80] and see Table 2).

### 7.2. The KDM1 and KDM5 Demethylases Active on Methylated H3K4

As with the demethylation of methylated H3K36, there are two families of demethylases which demethylate methylated H3K4 (see Figure 1). The two enzymes of the KDM1 family (KDM1A/LSD1 and KDM1B/LSD2) remove methyl groups from dimethylated and monomethylated H3K4 and the four KDM5 proteins (KDM5A, KDM5B, KDM5C, and KDM5D) demethylate H3K4me3 and H3K4me2. KDM5A and KDM5B can also demethylate H3K4me1 as determined by knock down studies and immunohistochemistry of transfected stained cells [137,138,139]. Although primarily demethylating H3K4me1/me2, LSD1 was reported to demethylate the repressive methylated H3K9mark in prostate cancer and thus enhance the function of the androgen receptor (AR) [140].

*LSD1 (KDM1A) and LSD2 (KDM1B):* LSD1 was the first histone demethylase to be identified, acting as a component of the CoREST histone deacetylase repressor complex [13]. LSD1 is also found in the NURD repressive complex (Mi-2/nucleosome remodelling and deacetylase complex) [141]. LSD2 is a homologue of LSD1 but is structurally very different and associates with different factors or complexes [142].

Expression of LSD1 is high in mouse and human ESCs, where it contributes to the maintenance of pluripotency [143,144]. Indeed, LSD1 is essential for development, and knockouts in mice are early embryonic lethals. Through its involvement with the CoREST complex, LSD1 can act as a transcriptional repressor and, functioning within the NURD complex, LSD1 can decommission enhancers of pluripotent genes during differentiation of mESCs [145]. Interaction with other components of the CoREST complex are required for LSD1 demethylase activity. There is an extensive literature covering the structure and biological functions of LSD1 and the reader is referred to a recent comprehensive review on this subject [146].

*The KDM5 demethylases:* The four KDM5 demethylases, KDM5A/B/C/D, can demethylate H3K4me3 and H3K4me2 in in vitro assays and KDM5A and KDM5B (but not KDM5C and D) can also demethylate H3K4me1 in transfected cells [137,138,139,147,148,149,150,151,152,153]. While demethylation of H3K4me3 at promoters is involved in transcriptional repression, KDM5A and B could decommission enhancers by removing the H3K4me1 mark, and this has been confirmed for KDM5B acting on the FGFR4 enhancer [154].

The catalytically active domain of KDM5 demethylases is divided into the JmjN and JmjC domains and sequences separating the two domains code for the ARID/BRIGHT DNA binding domain [155,156] and the first PHD domain (PHD1), which binds to unmethylated H3K4 [157,158,159]. Approaches using mutational analysis or specifically deleting the ARID and/or the PHD domains have given conflicting results regarding the requirement of the ARID domain for demethylase activity [137,138,139,160]. However, there is agreement that the PHD1 domain can influence the re-modelling of the catalytic core by binding to H3K4me0 [15,19,161]. Our own data from the ΔARID mouse expressing KDM5B with a deletion of the ARID domain, together with five amino acids from the JmjN domain (but with the PHD1 intact), show that this mutant has lost demethylase activity [85,86]. However, the fact that the ΔARID mouse is viable and fertile indicates that domains other than the demethylase activity are important in development and viability (See Table 2 and below). 

*Effects of KDM5 mutations affecting neural activity*: Mutations of KDM5 proteins are associated with intellectual disability and autism [151,153,162]. While *KDM5B* is found on chromosome 1 and *KDM5A* on chromosome 12, the *KDM5C* gene escapes silencing on the X chromosome and *KDM5D* is located on the Y chromosome. There has been some divergence of functions between KDM5C and KDM5D, but both proteins express KDM5 demethylase activity [163] and also play a role in cardiac function [164]. However, human males with mutations in *KDM5C* show cognitive abnormalities (X-linked intellectual disability: XLID) [153]. Clearly, the role of KDM5C in neural function is not rescued by WT KDM5D, even though it has demethylase activity [163]. In a mouse model expressing a demethylase null KDM5C protein, males show the behavioral defects seen in human males with XLID [152], suggesting that the demethylase activity of KDM5C plays a role in neural development. Furthermore, the dominance of mutations in the catalytic JmjC domain of *KDM5B* found in patients with ID also suggests that the KDM5 demethylase activity plays an important role in neural function [165]. However, both demethylase-dependent and -independent functions of KDM5 proteins may be involved in neural activity. It is important to ascertain the relevance of the demethylase activity of KDM5 proteins to neural function in the context of the development of inhibitors of this activity for potential therapeutic use.

Knocking out individual *KDM5* genes in mice has allowed their function to be investigated, but their effects on cognitive functions have only recently been examined. The first KO of KDM5A was reported to have only minor changes in phenotype [148] but a more recent KDM5A KO was found to have cognitive and physical disabilities [162]. Different phenotypes have been reported for KO of KDM5B in C57Bl/6 with Catchpole et al. reporting early embryonic lethality [85], and Albert et al. finding perinatal death due to respiratory failure, resulting from neurological abnormalities [166]. Since the ΔARID mouse, expressing a demethylase null mutant of KDM5B, is viable and fertile [85,86], it follows that functions other than demethylase activity are important in development. The ΔARID mouse provides an appropriate model for examining the effect of the loss of KDM5B enzyme activity on neural activity and behaviour.

*KDM5 protein interactions with repressor complexes:* Many studies have focused on the demethylase activity of the KDM5 enzymes, acting as transcriptional repressors, but other domains of the KDM5 and components of the complexes can be involved in this activity. KDM5B binds to HDAC1 and to Class II HDACs and the PHD1 and PHD2 domains in KDM5B are crucial for binding to HDAC4 [167]. Through binding to HDAC1, KDM5B binds to the repressive LSD1/NURD deacetylase complex and shows co-occupancy with the NuRD complex on chromatin [159,168]. KDM5A and KDM5C also bind to HDAC-containing complexes [169,170] and the activities of demethylation and acetylation are interlinked [171]. The combination of LSD1 and KDM5 proteins ensures the complete demethylation of H3K4 and, coupled with HDACs, this provides a very strong repressive signal.

The KDM5 demethylases can also bind to PRC2, suggesting coordinated removal of an active mark (H3K4me3/2) with the repressive action of Polycomb. In differentiation of mouse ES cells, interaction of the PRC2 complex with the KDM5A protein is required for repression of a significant number of Polycomb target genes [172] and a similar interaction of KDM5B with PRC2 is required to repress retinoic acid receptor target genes in the absence of retinoic acid [173].

## 8. Changes Associated with H3K4 Methylation in Cancer

Of the six Compass complexes that can methylate H3K4, KMT2A/MLL1, KMT2C, and KMT2D (MLL3 and 4) have been most widely implicated in cancer.

### 8.1. KMT2A/MLL1 and Development of BET Inhibitors

Largely due to the discovery of the translocations of KMT2A-producing fusion proteins in 70% of infant leukaemias and 10% of adult leukaemias [63,64], concentrated efforts have been applied to understanding the function of KMT2A in leukemogenesis. More than 80 partners have been identified as being linked in frame to a 5′ sequence of one allele of WT *KMT2A* to produce the oncogenic fusion proteins, designated as such by their ability to target the promoters of some oncogenes, as well as *HOX* genes [80,174,175]. The 5′ KMT2A sequence found in fusion proteins does not contain the catalytic SET domain but retains the ability to interact with PAFc (polymerase associated factor c) [176]. Importantly, most of the partners forming the 3′ end of the fusion proteins are components of or interact with components of the SEC complex (super-elongation complex) which contains pTEFb (positive elongation factor b).

The BET proteins (bromodomain and extra terminal BRD2,3 4), which bind to acetylated histones, also interact with PAFc and pTEFb and can recruit the fusion proteins to acetylated chromatin. BET inhibitors are now being evaluated for cancer therapy, not just in leukaemia, but for some solid cancers, in prostate cancer with androgen deprivation therapy, and in breast cancer, with fulvestrant hormone therapy [52]. Resistance can, however, develop to BET inhibitors in leukaemia patients [177] and other targets for drug development in leukaemia [174,178] are under investigation. The 5′ domain of KMT2A in fusion proteins retains the DNA binding site for menin and the leukemogenic activity of the fusion proteins is dependent on this interaction for recruitment to chromatin [63]. Moreover, menin is involved in the recruitment of the androgen receptor (AR) to target genes. Therefore, inhibitors of the menin interaction are being considered for therapy in leukaemic patients with KMT2A translocations and for patients with castration-resistant prostate cancer [52]. In the absence of translocations, in-frame partial tandem duplication (PTD) of exons 5 through to 11 occur in 4–7% of patients with AML [62], see Table 1.

### 8.2. Mutations in KMT2C and KMT2D

KMT2C (MML3) and KMT2D (MLL4) are among the most frequently mutated genes in cancer and the mutations inactivate function [56]. KMT2C mutations are found in common carcinomas with mutations in bladder cancer being over 20%. KMT2D mutations are also found in carcinomas, as well as in non-Hodgkin’s lymphoma and some neuro-ectodermal tumours [55]. Pre-clinical studies have provided data demonstrating the important role these methylases play as tumour suppressors and the crucial role they play in activation of enhancers ([18,179] and see Table 1).

Although mutations in the SET domain of *KMT2C* and *KMT2D* are common in cancer (>25% of mutations), mutations are also found in other domains, including the PHD domains. Oncogenic mutations in *KMT2C* in breast cancer occur at hotspots in PHD domains which, in WT KMT2C, interact with the tumour repressor BAP1 (BRACA1-associated protein 1). BAP1 recruits KMT2C to chromatin and opposes the action of Polycomb by removing the ubiquitin added to H2A119 by the PRC1 complex. [180]. Mutations in these hot spots block the interaction of KMT2A with BAP1 and many genes become aberrantly down-regulated by the Polycomb complexes.

### 8.3. LSD1 (KDM1A) in Cancer

LSD1 is over-expressed in many cancers and a range of inhibitors targeting LSD1 have been developed based on a small molecular weight compound (TCP tranylcypromine), originally approved for treatment of anxiety disorders, see Figure 3. TCP only moderately inhibits (irreversibly) the amine oxidase activity of LSD1. However, using TCP as a scaffold, modified structures have been developed carrying different covalent modifications which are more effective in inhibiting the demethylase function of LSD1 and the proliferation of cancer cells. TCP and its derivatives operate by covalently binding to FAD, located within the active site of the demethylase, and the effects are not reversible. Preclinical studies with these inhibitors of LSD1 indicate that the most sensitive cell lines were derived from AML or small cell lung cancer (SCLC) [181], and clinical trials with LSD1 demethylase inhibitors are ongoing for patients with these cancers [52,81]. However, preclinical studies indicate that other cancers could be appropriate targets for LSD1 inhibitors, particularly if used in combination with other therapies such as HDAC inhibitors [52,81].

*Oncogenic effects of LSD1(KDM1A) independent of the demethylase activity, see*Box 3*:* Important recent publications [82,182] have shown that in enhancing the growth of castration-resistant prostate cancer (CRPC cells), LSD1 activates a gene network dominated by cell cycle and DNA replication genes and that LSD1 ablation results in growth inhibition. This oncogenic effect is independent of the demethylase activity, and depends on the interaction with the ZNF217 protein, classified as an oncogene because of its increased expression in many cancers. An allosteric inhibitor has been developed which blocks the binding of ZNF217 to LSD1 and inhibits CRPC growth [82], (see Figure 3). Since ZNF217 can also form a complex with KDM5B, it is possible that interactions of KDM5B or other KDMs with ZNF217 occurs in other cancers [183].

LSD1 plays an important role in maintaining the oncogenic programme in leukaemias, showing a translocation of KMT2A and its inhibitors, which are in the clinic induce differentiation of the leukaemic blast cells [184,185,186]. However, pre-clinical studies with AML cell lines have shown that the action of the LSD1 inhibitors (developed to inhibit LSD1 demethylase activity) depends on inhibiting the interaction of LSD1 (in the CoREST complex) with the transcription factor GFI1B (or GF1) (see Figure 3). This disruption of the GFI1/LSD1 complex is independent of LSD1 demethylase activity, and leads to activation of expression of myeloid differentiation genes [187,188]. Vinyard et al. have used the technique of CRISPR–Cas9 mutagenesis to analyse the interaction of mutants of LSD1 with the small molecular weight inhibitors of LSD1 and drug resistance [189]. Their data confirm the crucial role of the GFI1B/LSD1 complex in AML and show that LSD1 mutants resistant to the GSK LSD1 inhibitor are enzymically inactive. CRISPR-suppressor scanning can detect differences in guidance RNAs and coding mutations selected with other inhibitors, giving new information on structure–activity relationships.

*LSD1/EZH2/HOTAIR:* The finding that the EZH2 Polycomb complex and the LSD1 complex can be associated by binding to different domains of the long non-coding RNA HOTAIR has added a new dimension to possible therapies based on this complex [190]. HOTAIR is over-expressed in many cancers, including breast and prostate carcinomas, and adult glioma [191,192], and the level of expression of HOTAIR determines to some degree the extent of LSD1/PRC2 binding. The expression of HOTAIR is very high in breast cancer metastases and already a small molecular weight inhibitor of the interaction of the EZH2 complex with the 5′ end of HOTAIR has been developed ([74] and see Polycomb section). In gliomas, expression of HOTAIR has been shown to be driven by the BET protein BRD4, and BET inhibitors can reduce expression of HOTAIR [192].

Box 3Targeting enzyme independent actions of LSD1.Of all the histone demethylases targeting methylated histones, progress in evaluating clinical efficacy is most advanced with LSD1 and the development of inhibitors for LSD1 has been directed to targeting the enzyme activity. However, pre-clinical studies show that in some cancers, an interaction of LSD1 with tran-scription factors, acting independently of the enzyme activity, is crucial for prolif-eration, and inhibitors of these interactions are under development (see Table 2 and Figure 3).

### 8.4. KDM5 Demethylases and Cancer

While it is abundantly clear that KDM5A, KDM5B, and KDM5C can be over expressed in multiple cancers, defining their individual and separate roles in oncogenesis presents a problem. There is an expansive literature describing preclinical studies with the KDM5 proteins, their known functions, and their expression in cancer (e.g., [15,16,17,193]). KDM5B, has been—and is—the most widely studied KDM5 protein in cancer, and several reviews covering KDM5B are also in print [194,195].

Since KDM5B was identified as being upregulated in breast and prostate cancer [137,138,149], this member of the KDM5 family has been widely studied in these cancers. In breast cancer, KDM5B is expressed most highly in the ER +ve subgroup, being classified as a luminal lineage-driving oncogene [196]. However increased expression of KDM5A, B, and C enzymes is seen in a wide range of cancers. Evidence that these enzymes can drive cancer cell proliferation comes from observations using cell lines, which show that levels of expression of the demethylase correlate with poor prognosis, and that knock-down of expression results in inhibition of cell growth in these cell lines [59,197,198,199]. KDM5 inhibitors which target the demethylase activity can be used to ask if the changes induced by KDM5 knock-down are demethylase-dependent [197].

*KDM5 proteins as tumour suppressors:* The KDM5C protein acts as a tumour suppressor in clear cell renal cell carcinomas (ccRCC), where the incidence in males-to-females is 2:1, and mutations in *KDM5C* are high [199]. KDM5C may also be involved in human papilloma (HPV) malignancies, since the E2 tumour suppressor protein recruits KDM5C to repress expression of the E6 and E7 oncoproteins [200]. KDM5D acts as a tumour suppressor in the prostate, where it interacts with the androgen receptor to down-regulate expression of AR target genes. In prostate cancer, downregulation of KDM5D is seen, thus enhancing expression of AR target genes [201,202].

KDM5B inhibits initiation of leukaemogenesis in leukaemias expressing MLL fusion proteins by reducing the high levels of H3K4me3 required by the leukaemic stem cells (LSCs) for proliferation. KDM5B shows reduced expression in the more differentiated leukaemic cells, so targeting KDM5B or other KDM5 proteins in leukaemia may not be an option [203].

*KDM5 proteins and drug tolerance:* One finding which suggests that the KDM5 demethylases could be cancer-therapeutic targets is that they play a significant role in the development of “drug tolerance”. KDM5A and KDM5B have been shown to be required for, and expressed by, a small population of reversibly drug-tolerant cells, (termed drug-tolerant persisters: DTP), which express stem cell markers. Although this work has been done largely with drugs used for lung cancer or melanoma, the principle has been found to apply to a wide range of cancer types and therapeutic drugs [204,205]. The reversibility of the drug-tolerant phenotype is evident when a cell line expanded from a drug tolerant clone (drug tolerant expanded persister DTEP) is cultured in the absence of the drug and becomes sensitive. This relates to the observation that patients who have acquired resistance to a specific drug can respond to a second treatment after a period of absence from drug treatment. DTPs have been found to show increased sensitivity to HDAC inhibitors, which do not inhibit the growth of parental cells. Combining an HDAC inhibitor with ablation of KDM5A effectively kills DTPs and DTEPs developing from several cell lines [204]. Moreover, combined treatment of cancer cell lines with the pan KDM5 inhibitor CPI 455 and HDAC inhibitors, or other targeting agents, also inhibits the development of drug tolerance ([206] and see below).

*KDM5 Inhibitors:* The association of enzymes that could remove methyl groups from the active mark H3K4me3 and their increased expression in cancer has led to the development of selective inhibitors of KDM5 demethylases, which have been applied pre-clinically to evaluate their possible use in cancer therapy. These inhibitors (eg CP70, CPI455, and pBIT) have selectivity for KDM5 versus other JmjC-dependent demethylases, but are not selective for individual members of the KDM5 family, which share highly conserved sequences [206,207,208,209,210], see Box 4. CPI455 has been found to inhibit cell growth and the development of drug tolerance in melanoma and NSLC cell lines [206], and to enhance the biological efficacy of a DNA methylase inhibitor (5-Aza-2′-deoxycytidine) and of HDAC inhibitors [206,210]. Inhibition of cell growth of multiple myeloma cell lines by CP70 has also been demonstrated.

Box 4KDM5 inhibitors targeting the demethylase activity requires further research.Currently, there are no KDM5 inhibitors in the clinic. Until more information is available on the functions of the individual KDM5 demethylases in specific cancers, and more inhibitors targeting individual members are available, clinical application may have to wait. It is possible that oncogenic effects which are independent of de-methylase activity may present opportunities for selective inhibition of the individ-ual KDM5 proteins. An important question is how important the functioning of the KDM5 demethylase activity in the adult brain is, in the context of inhibiting this function in cancer patients [86,152,211]. A comparison of behaviour in WT mice and the ΔARID mice expressing a demethylase-null KDM5B could give some answers.

## 9. Perspectives

There is considerable data arising from pre-clinical studies showing that disturbance of the methylation state of the histone lysines K4, K27, and K36 on histone 3 is a common event in cancer. These changes can occur as a result of mutations or translocations of the enzymes modifying the methylation state (Table 1), or to changes in their level of expression. While mutations in tumour suppressor genes, and changes of levels of expression, are found in a range of cancers, including the common carcinomas, other mutations and all translocations are associated with specific cancers, which are less common (see Table 1). Thus, inhibitors of the relevant KMTs or KDMs have been developed for potential clinical application in cancer therapy. However, only one methylase, EZH2 (KMT6A), and one demethylase, LSD1 (KDM1), are currently under extensive evaluation in the clinic for cancer therapy using inhibitors which target the enzyme activity.It is becoming clear that functions of the KMT and KDM proteins, which are independent of the enzyme activity, also play a role in development and in the changes occurring in some cancers (see Table 2 and Figure 3). However, pre-clinical research developing inhibitors for these histone methylases and demethylases has targeted the enzymatic activity. This approach makes it difficult to selectively inhibit specific homologues in families using the same chemical mechanism for enzymic activity. In evolution, the expansion of the number of genes with a common function from lower organisms to mammals (e.g., six homologues of the KMT2 methylase family, four dimethylating H3K36 and four of the KDM5 demethylases) is associated with increasing tissue complexity. This in turn emphasises the importance of cell phenotype in the expression and function of epigenetic factors. Obtaining more information regarding the differences in expression and function of the individual homologues in a family of KMTs or KDMs which may be seen in different tissues and in cancers developing from them is now a pressing area of research. Studies documenting the detailed structure of the homologues in a particular family could also give clues for identifying differential targeting. Taking this approach, inhibitors of the ASH1L methylase have recently been developed [117] which are selective in inhibiting ASH1L enzyme activity, but not the other methylases, which dimethylate H3K36. Such studies could lead to the development of not only enzymatic inhibitors specific for a particular homologue, but also for a new class of inhibitors targeting a function which is independent of the enzyme activity, and may be operative in specific cancers.

## Figures and Tables

**Figure 1 cells-11-01113-f001:**
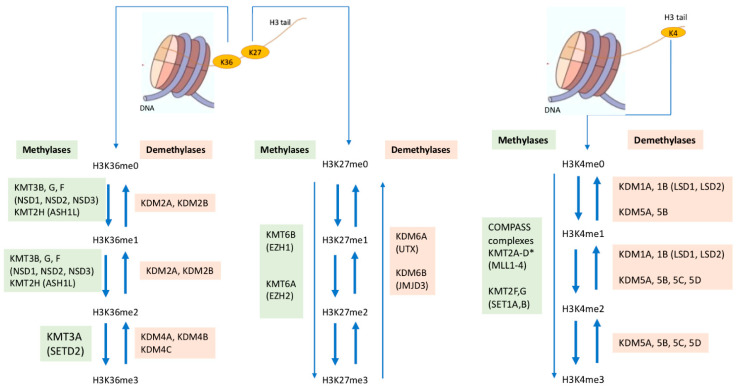
Action of the methylases and demethylases involved in the creation and removal of K36, K27, and K4 methyl marks on histone 3. Names in brackets are alternative names for the indicated enzymes. * KMT2A (MLL1) trimethylates a set of genes including *HOX* genes; KMT2B (MLL2) adds methyl groups to bivalent promoters in ESC; KMT2C (MLL3) and D (MLL4) can also add the monomethyl mark to H3K4 on active distal enhancers.

**Figure 2 cells-11-01113-f002:**
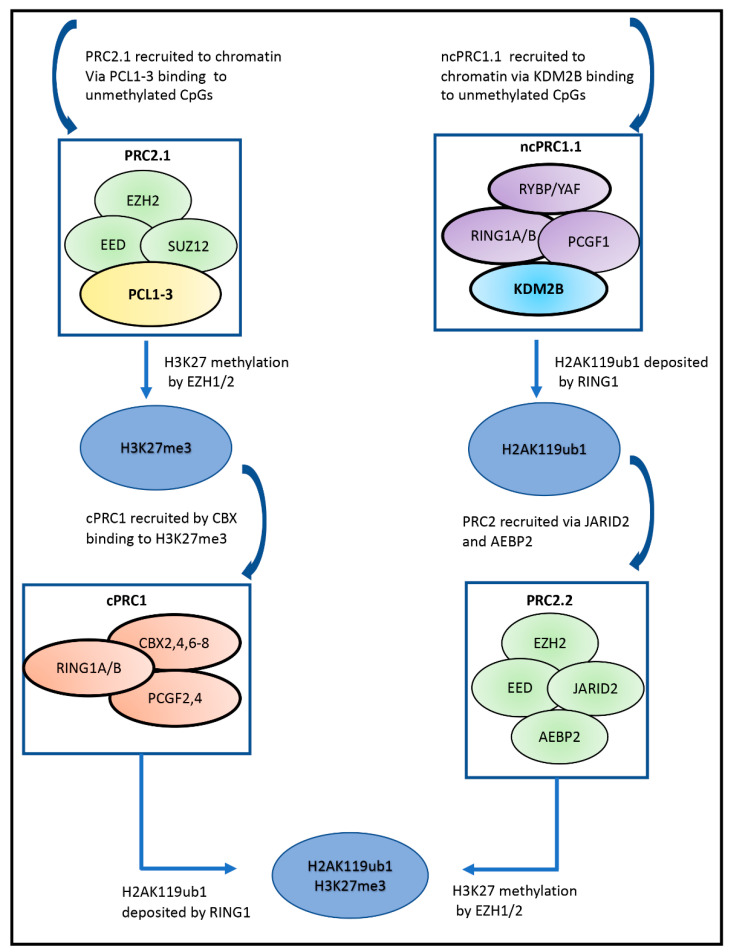
Formation of the K27me3 and H2AK119ub1 marks by components of the Polycomb complexes. Polycomb complexes cooperate to form the H3K27me3 and H2A119ub1 marks. (**Left-hand side**) PRC2.1 is recruited to chromatin by PCL1-3 binding to unmethylated CpGs, allowing EZH2 (KMT6A) to methylate H3K27. cPRC1 is then recruited via CBX binding to the H3K27me3 mark, allowing H2AK119 ubiquitination via the E3 ligase RING1A/B. (**Right-hand side**) The ncPRC1.1complex is recruited to chromatin by KDM2B binding to unmethylated CpGs, allowing ubiquitin to be deposited on H2A119 by RING1A/B. This allows PRC2.2 to be recruited via Jarid1B and AEBP2 and the formation of the H3K27me3 mark by EZH2.

**Figure 3 cells-11-01113-f003:**
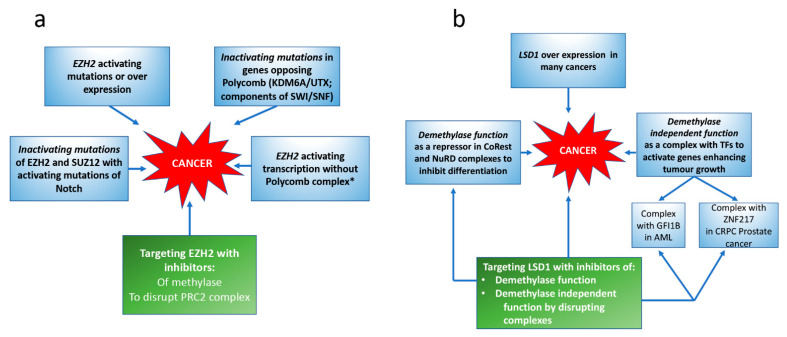
Involvement of EZH2 and LSD1 in cancer and their use as potential therapeutic targets. (**a**) Involvement of EZH2 in cancer. (**b**) Involvement of LSD1 in cancer. Blue boxes indicate the different ways EZH2 or LSD1 are implicated in cancer; green boxes, potential targeting approaches. * Governed by methylase dependent and independent pathways in CRPC.

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
