# Peer review of "Histone Methylases and Demethylases Regulating Antagonistic Methyl Marks: Changes Occurring in Cancer"

_cells, 2022, doi:10.3390/cells11071113_

Round 1

Reviewer 1 Report

The manuscript by Papadimitriou and Burchell review histone methylases (and demethylases) belonging or related to the Polycomb and Trithorax subset of epigenetic regulators. Thus, methylation of H3 tails at K4 and K27, dependent on Trithorax and Polycomb complexes, respectively, is central, while a subset of K36 methylation relies on the activity of Trithorax complexes. This, by itself, represents a large volume of information of which much is not yet understood as to be practical in mechanistic description of the underlying functions. On top of this, the authors include information on the demethylases relevant to these histone modifications. Altogether this makes for a challenging task when considered in a cancer context where phenomenology is uncontrolled. 

Perhaps because of this, the reviewed material reads, fundamentally, as a huge catalog of observations of difficult digestion. I wonder whether an introductory part would have been useful describing the cross-talks between these histone modifications, so that the system appears as interlinked as, in fact, is. Likewise, only at the end is mentioned the non-catalytic contributions of the methyltransferases involved. Also not clear: that histone modifications, per se, are not necessarily instructive regarding transcriptional outcome and that functions of chromatin regulators are known to be cell type and developmental dependent; therefore, combining evidence from a variety of biological models does not help reading. The same when information pertaining to the mentioned histone modifications are carried out by non-Trithorax components (much of what it is included under the H3K36 methylation section) or non-histone postranslational modifications dependent on PRC2 subunit EZH2. More: demethylase LSD1, is not a Polycomb/Trithorax component.

Given the cross-talk between modifications at these lysines in histone H3 it would have made sense to consider oncohistones together rather than separated in a modification-dependent manner.

I wonder whether more extensive, heavier, tables may convey much of the information in the text that reads as a catalog. Try to keep most of the text for what can be used as narrative, specifying what it could be expected or the questions not yet studied. A small, separated section about therapeutic avenues (and their limitations) probably would be useful. The main figures are good, those in the supplementary material add little given that the myriad of domains and subunits are not reflected in functional descriptions (not known) and then appear more as a list of complex protein structures.

Given that much of the information is already available in other reviews, it is, perhaps, a summarized version what it would be useful rather than an attempt to become comprehensive, thus improving the review. Finally, what imbalance the tittle refers to? 

Author Response

Replies to Reviewer 1

 Reviewer’s comment:  The manuscript by Papadimitriou and Burchell review histone methylases (and demethylases) belonging or related to the Polycomb and Trithorax subset of epigenetic regulators. Thus, methylation of H3 tails at K4 and K27, dependent on Trithorax and Polycomb complexes, respectively, is central, while a subset of K36 methylation relies on the activity of Trithorax complexes. This, by itself, represents a large volume of information of which much is not yet understood as to be practical in mechanistic description of the underlying functions. On top of this, the authors include information on the demethylases relevant to these histone modifications. Altogether this makes for a challenging task when considered in a cancer context where phenomenology is uncontrolled. 

 Authors’ reply

We realise that the review covers a wide area, reporting on the proteins which can affect the methylation status of antagonistic histone methyl marks.  We feel however that covering the current state of knowledge in this area is important and requires inclusion of the histone demethylases: to help the reader we have used small boxes to summarise each section. Including the KDMs and the methylation of H3K36 distinguishes this review from others covering only methylases or only demethylases or the classic Polycomb /Trithorax view considering only the K27 and K4 methylation states. Incorporating the methylases and demethylases for H3K36 has allowed a discussion of the important interaction of the KDM2B demethylase with a Polycomb complex.(see new lines 69-70) We have now emphasised this in the Introduction  (see new lines 63- 74). We have also made some adjustments in the text (deletions) in reply to the authors comments on the narrative style of the text (See below).

Reviewer’s comment: I wonder whether an introductory part would have been useful describing the cross-talks between these histone modifications, so that the system appears as interlinked as, in fact, is. Likewise, only at the end is mentioned the non-catalytic contributions of the methyltransferases involved.

 Authors’ reply

We have taken on board this comment. At the end of the introduction there is a box with text where we have now introduced the possibility of enzyme independent functions.  We have included in this box a reference to an example of how a KDM and KMT can interact to activate enhancers in a manner which does not involve the catalytic activity of the KDM (reference 18). We have also now referred to this interaction in the Polycomb section (lines  193-196 in new pdf version)

Reviewer’s comment: Also not clear: that histone modifications, per se, are not necessarily instructive regarding transcriptional outcome and that functions of chromatin regulators are known to be cell type and developmental dependent; therefore, combining evidence from a variety of biological models does not help reading”

Authors’ reply

We have discussed the importance of cell phenotype in defining the epigenome  in the Perspectives section. Moreover one of the reasons for reporting on individual  cancers which exhibit a specific change in the epigenome is to avoid reporting the change as modified in “Cancer” in a general way, and to emphasise the importance of cell phenotype. This is particularly important for those changes which are driving cancer proliferation.  Moreover, listing the cancers separately which over express a specific epigenetic factor has allowed reporting on the different transcriptional responses in the different cancers (for an example, see effects of over expression of KDM2B in ALL, pancreatic cancer and synovial sarcomas lines, under the heading “KDM2B over-expression in cancer” on page 12).  

Reviewer’s comment: The same when information pertaining to the mentioned histone modifications are carried out by non-Trithorax components (much of what it is included under the H3K36 methylation section) or non-histone postranslational modifications dependent on PRC2 subunit EZH2. More: demethylase LSD1, is not a Polycomb/Trithorax component.

Authors’ reply.

  • The reason we  included  the studies on methylation of H3K36 is because the marks produced can oppose Polycomb. We have mentioned this in the Introduction, and indicated that of the methylases, which methylate H3K36, only the ASH1L  methylase operates as a complex and is recognised as Trithorax   (lines 63-64)The involvement of the demethylase KDM2B in a non-canonical Polycomb complex is important in the studies defining recruitment of the Polycomb to chromatin.
  • The inclusion of the action of EZH2, which does not require it to be in the complex, is reported because of the therapeutic targeting of EZH2 in Cancer. Also, although none of the demethylases can be classified as Trithorax, (although they are found in Drosophila) drugs targeting LSD1 in Cancer are also in the clinic.
  • To clarify that the review is not limited to the Polycomb/trithorax antagonistic system originally defined in Drosophila, we have changed the title to “Histone methylases and demethylases regulating antagonistic methyl marks: changes occurring in Cancer’

Reviewer’s comment: Given the cross-talk between modifications at these lysines in histone H3 it would have made sense to consider oncohistones together rather than separated in a modification-dependent manner.

Authors’ reply

We chose the modification of methylation at specific sites acting antagonistically, as a guide to identify the proteins which had the potential to influence the methylation (methylases and demethylates) at these sites.  We have then considered the possible importance of enzyme independent functions of these proteins in carcinogenesis and in the development of drugs for Cancer Therapy. This is particularly relevant to the current trend to develop drugs directed to the enzymic activity of these proteins.

 Reviewer’s comment:   I wonder whether more extensive, heavier, tables may convey much of the information in the text that reads as a catalog. Try to keep most of the text for what can be used as narrative, specifying what it could be expected or the questions not yet studied.

Authors’ reply

  • We explained above that listing specific cancers where a documented epigentic change has occurred emphasises the context of cell phenotype. It also allows important details to be mentioned rather than condensed into a very busy Table. We have however, added some more details to Table 1 to take on board your comment and to reply to the comment from another reviewer who wanted inclusion of other mutations and translocations. Where possible we have tried to modify the text to have the narrative form.
  • We have addressed the comment “text that reads as a catalog” by rewriting parts of the text and deleting some of the “lists” that were present in the text. Under the heading “Methylases in COMPASS COMPLEXES (Complex Proteins Associated with SET1)” (page  14) we have deleted the list of KDM2s and their activities and condensed some of this information into the legend for figure 1. We have also rationalised the KDM6A and 6B demethylases in carciomas into a single narrative. Other redundant sections been  removed as indicated by tracking.

Reviewer’s comment: A small, separated section about therapeutic avenues (and their limitations) probably would be useful.

Authors’ reply

 Figure 3 was included to summarise the aspects which need to be considered for the two proteins (EZH2 and LSD1) which are being extensively evaluated for Cancer Therapy. For these factors, the inhibitors available have been discussed in the text

Reviewer’s comment: The main figures are good, those in the supplementary material add little given that the myriad of domains and subunits are not reflected in functional descriptions (not known) and then appear more as a list of complex protein structures.

Authors’ reply

 We have now deleted the figures in the Appendix and any reference to them in the text.

Reviewer 2 Report

The manuscript is a bit oversimplified in the description of the cancer related mutations/abnormalities in methylases and demethylases. Many important driver events are missing, e.g., translocation of NSD2 (MMSET, t(4;14)) in  15%-20% Multiple myeloma and hotspot mutation (E1099K) in pediatric ALL.

Table1: again, the table is far from comprehensive. Please also double check the translocation rate of MLL/KMT2 in pediatric leukemia (ALL or AML?), usually shouldn’t reach 70%. Translation of MLL and partially tandem duplication (MLL-PTD) events are also frequently observed in adult AML. SETD2 mutations also observed in some lymphoma.

Author Response

Reply to reviewer 2

Reviewer’s comment: The manuscript is a bit oversimplified in the description of the cancer related mutations/abnormalities in methylases and demethylases. Many important driver events are missing, e.g., translocation of NSD2 (MMSET, t(4;14)) in  15%-20% Multiple myeloma and hotspot mutation (E1099K) in pediatric ALL.

Authors’ Reply:

 We thank the reviewer for pointing out these omissions which we have now rectified. We have added the NSD2 translocation t(4:14) to Table 1 with the appropriate references and also included this in the text under the section headed “ Translocations and mutations of NSD (KMT3B, G, F) methylases”.

The glutamic acid to lysine mutation found in NSD2 has also been added to Table 1 (with references) and is included in the text under the heading mentioned above.

Reviewer’s comment: Table1: again, the table is far from comprehensive. Please also double check the translocation rate of MLL/KMT2 in pediatric leukemia (ALL or AML?), usually shouldn’t reach 70%. Translation of MLL and partially tandem duplication (MLL-PTD) events are also frequently observed in adult AML. SETD2 mutations also observed in some lymphoma.

Authors’ Reply

 We have checked the translocation of KMT2A and can confirm it is 70% in infant leukemias (both ALL and AML) although less frequent in leukemias from older children (see reference 167, Krivtsov and Armstrong 2007). We have now replaced the word “childhood” with “infant” to remove any ambiguity and included that ALL and AML can be involved.

Although the papers refer to MLL and no specific KMT, they quote the chromosome position of the gene to be 11q23. This is the position of KMT2A and does not correlate with the position of other KMT2s.

We have now included a reference to partially tandem duplication (MLL-PTD) events in leukemi  (Dorrance et al 2006, reference 174) and referred to this reference in the text and in Table 1.

We have also added to table 1 that SETD2 mutations occur in 32% of enteropathy-associated T cell lymphomas and included a reference to this in the text under the heading “SETD2 (KMT3A): A tumour suppressor”.

Round 2

Reviewer 1 Report

The authors have made an effort to accommodate the previous criticisms.

Although, in essence, the manuscript remains similar, a more extensive rewriting is probably not worth doing. Readers, particularly those new to the field, will find a large collection of informative bits gathered in one place, which will be useful as a starting point. Also, I believe literature review work, provided accuracy and representation, has a large component of personal preferences in its style.

This manuscript is a resubmission of an earlier submission. The following is a list of the peer review reports and author responses from that submission.

Round 1

Reviewer 1 Report

There are lots of problems with this article. If written English is not right then the whole message is distorted. The first sentence has a typo. At places instead of PRC2, the authors have mentioned PCR complex.

  1. The authors should indicate the number of amino acids in every methyltransferase.
  2.  The authors should present a model diagram for every disease condition mentioned.
  3. Write about the cross-talks between the methylation marks like H3K27me3 and H3K36me3.

Reviewer 2 Report

Taylor-Papadimitriou and Burchell present a review with the laudable aim to summarize our current knowledge about Trithorax complexes, Polycomb repressive complexes, the methylation marks on H3K4 and H3K27 set by these complexes and the demethylases known to be able to remove the same methylation marks. Furthermore, they present differences in relation with cancer. This is an extensive scope and cannot be comprehensive. While I would like to thank the authors for the strong effort, I feel that with 27 text pages (without references) the review is too long. Instead of a one-by-one discussion, I would have preferred to read a synthesis and to find tables summarizing details (such as reported changes in cancer with references). The authors could further consider condensing information in figures. 

My main suggestions are:
- to increase the level of synthesis, 
- to focus more on the chosen topic, maybe reducing the scope, and 
- to shorten, shorten, shorten.

Figure 3 and 6. I understand what the authors wish to do. But would a phylogeny be scientifically more correct. Alternatively, please consider if cartoons for erasers and readers for H3K4me3 and H3K27me3 would be more intuitive to read. Are the yeast and Drosophila proteins relevant for this review not discussing evolution?

Figure 2 and 4A. Please check the nucleosome structure. It seems that H3 is drawn on the wrong side (should be next to DNA entry).

Figure 2 and 4B. Please consider if the domain level of detail is helpful. The text does not really discuss the biochemical properties of the shown domains. Some domains are not mentioned at all.

Figure 5. Again is this level of biochemical detail helpful for a cancer-oriented review?